# Attachment to Mother and Father, Sleep, and Well-Being in Late Middle Childhood

**DOI:** 10.3390/ijerph20043399

**Published:** 2023-02-15

**Authors:** Catarina Perpétuo, Mona El-Sheikh, Eva Diniz, Manuela Veríssimo

**Affiliations:** 1William James Center for Research, ISPA, 1100-287 Lisbon, Portugal; 2Department of Human Development and Family Studies, Auburn University, Auburn, AL 36849, USA

**Keywords:** attachment, secure base support, safe-haven support, sleep quality, well-being, late middle childhood

## Abstract

The security of attachment has been related to several advantageous developmental outcomes, such as good sleep quality and higher well-being indicators. However, few studies concern the associations between attachment dimensions to both parents, sleep, and well-being in late middle childhood. Our study aims to expand knowledge in this area, clarifying the above-mentioned associations by considering the secure base and safe haven dimensions of attachment. We also investigate the role of sleep as a mediator of the relationship between attachment and well-being. The 258 participants (49.2% girls, mean age = 11.19, *SD* = 0.85) completed self-report questionnaires regarding attachment (KSS), sleep (SSR), and well-being (CHIP-CE). The results show significant associations between attachment to both parents (0.40 ** ≤ *r* ≤ 0.61 **) and between attachment security, sleep (−0.21 ** ≤ *r* ≤ −0.35 **) and child well-being (0.42 ** ≤ *r* ≤ 0.47 **). Besides, sleep quality partially mediated the relations between all attachment dimensions to both parents and well-being. The results are discussed in light of attachment theory, focusing on the comparison between attachment to mother and father as a valid framework to unravel differences in child well-being, with sleep as a process that can help to explain the mechanisms through which attachment security enables subjective perceptions of well-being.

## 1. Introduction

Child well-being is an umbrella term covering a sense of subjective satisfaction with both general and specific domains of life [1,2], such as social adaptation and mental/psychological and physical health. Considering the implications of well-being in child development (e.g., health, longevity, academic performance; [3,4,5,6]), the concept definition and operationalization is a priority in child development research. However, the multidimensional nature of the construct makes it difficult to measure and operationalize, allowing for different conceptualizations [7]. While some authors define it in terms of a latent construct measured, for example, by indexes of mental health (e.g., [8]) or psychological functioning (e.g., [9]), others conceptualize it in broader terms to include physical, social, and environmental aspects (e.g., [10]). However, more recent studies focused on holistic approaches, considering the interaction between different elements and their influence on subjective perceptions of well-being [11]. Among these elements is the capacity for experiencing positive emotions, social interactions, school adjustment [12,13], and absence of mental distress, for example [14,15]. In fact, the inherently subjective nature of well-being has led the authors to privilege self-report measurement approaches from middle childhood [16,17,18,19]). In our study, we relied on a well-established measure of well-being (CHIP-CE; [20] Portuguese version [21]) that assessed satisfaction, self-esteem, physical symptoms, mental health, academic achievement, and resilience, among others.

Late middle childhood or preadolescence (10–12 years) is a developmental period spanned by multiple changes that render children particularly vulnerable to stressors (e.g., [22]), potentially altering their own subjective perceptions of well-being. Accordingly, decreases in the subjective perception of well-being have been reported as the child enters preadolescence (e.g., [23,24,25]). At a bodily level, anatomic and physiological transformations accompanying the onset of puberty are triggered by neurochemical and hormonal processes that translate, for example, in changes in sleep cycles (e.g., [26,27]), self-esteem (e.g., [28,29,30]), and mood (e.g., [31]). In a social-relational sphere, the child’s social world expands (e.g., [32,33]), and a sense of autonomy from the primary figures increases, potentially altering attachment relationships with the parents. As dependency decreases and exploration of the environment takes a central role, the parents may need to adapt, balancing their control and guidance and renegotiating the mutually acceptable levels of autonomy and relatedness in the relationship [34,35,36,37]. Researchers agree that the relationships with the parents, often operationalized in terms of attachment relationships, are a determinant factor in predicting a child’s well-being during late middle childhood and future adjustment [38,39,40,41]. 

### 1.1. Attachment to Parents and Child Well-Being

Attachment theory has a highly relevant and well-validated framework explaining individual differences in adjustment across the lifespan (e.g., [42,43]) as a determinant factor in children’s physical/mental health and well-being [43,44,45]. Empirical evidence suggests that attachment security is associated with fewer psychiatric symptoms [46,47] and fewer adjustment, behavioral, internalizing, and externalizing problems [47,48,49,50]. In a study with early adolescents and adolescents, attachment security was associated with a decreased vulnerability to negative affect and to feelings of loneliness [51]. Regarding academic and peer competence, attachment security in late middle school children was associated with better scholastic, emotional, and total adjustment to school [52], socioemotional competence [53], emotional understanding [54], and to a better ability to tolerate frustration deriving from academic tasks without becoming overwhelmed [55]. Otherwise, insecure attachment is associated with dysregulation of the stress response [56], physiological hyperarousal [57], less self-regulatory abilities [58], and to a non-specific vulnerability to stress together with maladaptive strategies to regulate dysphoric affect [59]. When taken together, attachment insecurity is negatively associated with multiple domains of well-being.

The competence hypothesis of attachment theory offers a comprehensive framework for the relations between attachment and well-being, suggesting that a secure relationship with primary caregivers places the child on a more positive developmental trajectory [60]. This could impact a child’s well-being in at least three ways. First, securely attached children form more positive relationships with peers, cooperate more with adults, and feel more confident about themselves, which frees up their internal resources for a confident exploration of the environment, one of the main developmental tasks of preadolescence [60,61,62]. Second, because of their secure internalized models of relationships and of others, securely attached children are better able to acknowledge stress and trust others enough to seek help when needed, then providing a sense of well-being. Last, internalized experiences of dyadic regulation allow the child to self-regulate emotions more effectively, impacting the way in which life events are appraised.

Traditionally, developmental research has focused more on the role of the mother in child development, while the importance of the father has consistently been shown empirically (e.g., [63,64,65,66,67]). However, few studies have been dedicated to understanding the role of child–father attachment in child development beyond the early years (e.g., [68,69]). Although the security of attachment to mothers and fathers has shown to be significantly related (e.g., [70]), it is also true that children can be securely attached to one parent and not to the other (e.g., [71,72]) and that attachment to mothers and fathers can play distinct roles. For example, attachment to the mother seems to play a relevant role in emotional understanding, as well as in the integration of positive and negative feelings [73], in concordance with the attachment safe haven support function. In turn, attachment to the father seems to play a major role in supporting the child’s exploration of the environment [74], in line with the attachment secure base support function [75,76]. Therefore, we relied on an instrument capable of distinguishing attachment to mother and to father in the secure base and safe haven functions [77] to study their independent contributions to child well-being. In preadolescence, the exploration of the environment is one major developmental task. However, the needs for close emotional connectedness are still very important and may involve a renegotiation of the mutually accepted levels of proximity in the dyads [61].

### 1.2. Sleep as a Mediator of Relations between Attachment and Well-Being

Sleep is a complex bioregulatory system that is influenced by biological, familial, and broader societal factors [78,79]. There is a recognized need in the field to examine sleep as a mediator of the associations between family relationships and child well-being [80,81]. Since attachment serves self-regulatory functions, advancing research in this direction may extend understanding of the role of relationship factors in child development, particularly in health and well-being. Specifically, it would be important to understand how attachment relationships affect bioregulatory processes, such as sleep, and to what extent this could configure a mechanism through which attachment relationships impact well-being.

Using a well-established measure (SSR; [82]; Portuguese version [83]), the present study assesses multiple dimensions of sleep and its context through self-report: consistency in the timing of sleep (e.g., falling asleep around the same time each night), factors of the sleep environment (sleeping alone or co-sleeping with family members), sleep satisfaction (feeling that one obtains enough sleep), and daytime sleepiness [82]. Based on common practice [84], a total score was used and is generally thought to reflect sleep quality.

Poor sleep quality is prevalent and confers risk for youths’ well-being, mental health, and cognitive and academic functioning [85,86,87,88]. Importantly, both objective and self-reported sleep quality have significant consequences for adolescent outcomes, even beyond those of sleep duration. For example, poor sleep quality was associated with worse adjustment for adolescents and adults, even for those who had longer sleep duration [89]. Additionally, when considered with other sleep parameters, self-reported sleep quality has emerged as a robust predictor of physical and mental health [90]. When measured appropriately, health takes into account not only the absence of disease but the presence of wellness [91]. Sleep health, by extension, provides an opportunity for us to demonstrate what happens when sleep goes “right”—that is, to enumerate the positive outcomes, such as psychological well-being, that may be accrued from sufficient, high-quality sleep among youth [91,92].

The associations between sleep and multiple facets of adjustment in preadolescence are well-established in the literature, indicating that good sleep quality is linked to many developmental outcomes. Specifically, age-appropriate sleep duration and better sleep quality are associated with higher levels of well-being, measured as psychological attitudes towards life, future, recent events, family life and friendships, both cross-sectionally and 6 months later [92,93]. Conversely, shorter sleep durations predict lower self-esteem [94], mental health problems [95,96], internalizing and externalizing problems [97,98], risky behaviors [99], somatic complaints [100,101] and poorer academic functioning [102,103]. Even minor sleep disturbances are reflected in poorer well-being, defined as health-related quality of life in physical, emotional, social, and school domains in children between 10 and 11 years old [58].

Among the factors that can influence sleep quality, those associated with family processes are considered crucial [104], including children’s emotional security associated with parents’ marital conflict [105]. Attachment security to parents has been considered one of the factors that can impact sleep quality throughout life [106,107]. According to Dahl’s framework [108], sleep and vigilance are opposing processes. While the ability to relax is required to fall asleep, one must be awake to maintain awareness of threats/dangers. It follows that the high intensity of negative affect provoked by feelings of worry, characteristic of insecurely attached children, such as fears of being alone with no reliable others to provide care when needed, may compromise the ability to relax at night, therefore translating into problems falling and staying asleep. Conversely, attachment security would create a state of inner safety that would facilitate sleep. These theoretical claims have found empirical support in infancy, childhood, and adulthood [106,109]. However, only a few studies have investigated the relationship between attachment and sleep in middle childhood [105,110]. One of the studies found a negative association between attachment to the mother and self-reported sleep problems in children aged between 8 and 9 years old. However, sleep problems did not mediate the association between child–mother attachment and academic achievement [110]. The other study found that greater child–mother security predicted decreased daytime sleepiness in children and fewer sleep problems in boys, while child–father attachment security predicted increased sleep duration for girls from 3rd to 5th grade [105].

### 1.3. Study Aims

Although attachment security has been shown to play a role in sleep quality and in several dimensions of well-being, less is known about the relations between attachment security to both parents, sleep, and well-being in late middle childhood. Hence, the first goal of the present study is to clarify the referred associations. In addition to examining direct effects among these constructs, for the second goal, we hypothesized that sleep quality serves as a mediator of relations between attachment security to both parent and child well-being.

## 2. Materials and Methods

### 2.1. Participants

A sample of 258 fifth and sixth graders (49.2% girls, mean age = 11.19, *SD* = 0.85) was recruited from two public schools in the Lisbon area (Portugal), integrating a larger project about sleep and socio-emotional development. From 1200 letters initially sent to the families of children attending these schools, 477 agreed to participate in our study. However, due to overlapping school activities and children missing classes, the final sample of 258 corresponds to participants who had complete data for variables of interest in this study.

Most of the children’s parents were married to each other (73.5% vs. 26.5% of children with divorced/separated parents) and had full-time working parents (86.7% of the mothers worked full-time vs. 95.6% of the fathers). The majority of the children had one (36.2%) or two (28.3%) siblings, while 15.7% had three or more, and 19.7% were the only child. Maternal age varied between 26 and 57 years (*M* = 41.21, *SD* = 5.42), and paternal age varied between 28 and 69 (*M* = 43.34, *SD* = 6.17). Regarding maternal education, 41.6% completed a college degree, whereas 29.8% finalized 12th grade and 28.6% achieved 11th grade or less.

### 2.2. Procedure

We contacted the director of a preparatory public school in the area of Lisbon in March of 2018, and after a first meeting, we were allowed to contact the parents of the children who attended the classes ministered by the teachers who agreed to collaborate in our study. A cover letter of the project and the consent form were sent to the parents, explaining that their child’s participation would consist of answering some questionnaires related to sleep habits and child development in a classroom context. Data were collected between April and June of 2018, and each child took three weeks to complete our questionnaires in 45-minute weekly sessions. During the first week, children reported on their attachment relationships with both parents (KSS), and in the following weeks, they answered questionnaires about their sleep quality (SSR) and general health and well-being (CHIP-CE). The decision to extend the data collection for three weeks was meant to help the children read and answer the questions carefully without feeling overwhelmed.

### 2.3. Instruments

#### 2.3.1. Kerns Security Scale (KSS; [77])

This scale assesses children’s perceptions of parent–child attachment across two dimensions—Secure Base and Safe Haven Support—and it has been frequently used with 9–14-year-old children (see [77,111]). We used a previously adapted and validated version of the scale for the Portuguese population [112]. Both dimensions refer to both parents, resulting in four final scores: maternal safe haven support (SHS_M, Cronbach- α = 0.80), paternal safe haven support (SHS_F, Cronbach- α = 0.84), maternal secure base support (SBS_M, Cronbach- α = 0.76), and paternal secure base support (SBS_F, Cronbach- α = 0.72).

For each question, the child is presented with two different types of children and then has to decide which is more similar to him or her, e.g., “Some kids wish their mom would help them more with their problems BUT other kids think their mom helps them enough”. After picking which kids they are more like, the child specifies whether they are “sort of like” or “really like” the child in the question. The Safe Haven Support subscale (SHS; 14 items) evaluates open communication about needs and emotions and whether a child trusts in her/his parent(s) to provide protection and emotional care in times of distress, for example, “going to a parent when upset”. The Secure Base Support subscale (SBS; 6 items) assesses parents’ encouragement and support of exploration and decision-making, for example, “encouraging the child to be themselves or to try new things”. Each item is measured on a 4-point scale, resulting in average scores for SHS and SBS for both mother and father (higher scores reflecting greater security).

#### 2.3.2. Child Health and Illness Profile—Child Edition (CHIP-CE; [20,21])

Children’s reports of health and well-being were obtained through their responses to CHIP-CE, a self-report 5-point Likert (1—Never, 5—Always) scale designed for children aged from 6 to 11 years old. The questionnaire assesses the following domains: Satisfaction (9 items, α = 0.87), describing the child’s assessment of his or her well-being and self-esteem; Comfort (12 items, α = 0.74), assessing the degree to which physical and emotional symptoms and their associated activity limitations are endorsed by the child; Resilience (8 items, α = 0.73) characterizes the child’s states and behaviors that are likely to enhance future health; Risk avoidance (8 items, α = 0.77) is the child’s perception about how often he/she engages in behaviors that may be a risk to future health or development; Achievement (8 items, α = 0.77) addresses how well the child feels he/she performs both academically and socially with peers. For each item, two cartoon illustrations that depict the appropriate extreme state of health (e.g., for the item In the last four weeks, how many times did you cry a lot?, a child cartoon depicting a neutral face was placed near option 1—Never, and a crying cartoon near option 5—Always) are presented, and five possible response circles are given, graduated in size to indicate increasing/decreasing frequency or amount, with item wording placed beneath. The global score (alpha = 0.91) is obtained by adding the item scores for each of the five subscales, and higher scores indicate better health and well-being.

#### 2.3.3. Sleep Self Report (SSR; [82,83])

Children completed the SSR, a one-week retrospective questionnaire used to assess the subjective perception of multiple sleep dimensions, collectively referred to as quality. The questionnaire comprises three initial questions that are not considered in the scale global score (1. Who establishes sleep schedules at home?; 2. Do you think you have a sleep problem?; 3. Do you like sleeping?). For the next questions, children are asked about the frequency of occurrence of events related to different sleep domains—such as consistency of bedtime, difficulty initiating or maintaining sleep, satisfaction with their sleep, and daytime sleepiness (α = 0.70). The SSR’s 23 items are answered on a three-point Likert scale: 1 (“Rarely”; 0–1 times a week), 2 (“Sometimes”; 2–4 times a week) or 3 (“Usually”; 5–7 times a week), with higher scores indicating lower sleep quality.

## 3. Results

### 3.1. Descriptive Statistics

Table 1 depicts the mean scores for the whole sample and by gender for primary study variables. We started by exploring the perceptions of secure base and safe haven dimensions of attachment to mother and father. Overall, children in our sample reported secure attachment both to mother and father (mean scores vary between 3.22 (SD = 0.58) in a safe haven to father and 3.47 (SD = 0.47) in a secure base to mother). However, we found significant differences in secure base and safe haven scores between mothers (M_SBS_ = 3.47, SD = 0.47, M_SHS_ = 3.44, SD = 0.45) and fathers (M_SBS_ = 3.38, SD = 0.54, M_SHS_ = 3.22, SD = 0.58), meaning that children tended to attribute higher secure attachment scores to their mothers, reflected in both the secure base score (t (254) = 3.06, *p* = 0.002) and the safe haven score (t (254) = 7.40, *p* = 0.000). We also found that children attributed significantly higher scores of secure base support than a safe haven to the father (t (254) = 5.70, *p* = 0.000). However, we did not find differences between scores attributed to the mothers (t (257) = 1.29, *p* = 0.199). Regarding gender differences in attachment security, boys perceive themselves as receiving greater secure base support (M = 3.58) and safe haven support (M = 3.51) from their mothers than the girls (M = 3.35, SD = 0.04 and M = 3.35, SD = 0.04, respectively, F (1, 256) = 14.24, *p* = 0.000, eta = 0.05; F (1, 256) = 9.39, *p* = 0.002, eta = 0.04). No differences were found in how girls and boys reported on secure paternal base (F (1, 253) = 2.77, *p* = 0.097) and safe haven support (F (1, 253) = 0.01, *p* = 0.912).

### 3.2. Partial Correlations among the Study Variables

Partial correlation coefficients between the variables are displayed in Table 2. Attachment dimensions correlated significantly with each other, meaning that children who perceive one parent as a source of secure base support tend to classify him/her also as a safe haven in times of distress (r = 0.64, *p* < 0.05 for the mother, r = 0.68, *p* < 0.05 for the father) and to classify the other parent accordingly (r = 0.58, *p* < 0.05 for the secure base support between the parents and r = 0.61, *p* < 0.05 for the Safe Haven Support between the parents). Further, the more the children perceive their parents as secure base support and as a safe haven in times of distress, the fewer sleep problems are reported (the correlation coefficient varies between r = −0.21, *p* < 0.05 for maternal secure base support and r = −0.35, *p* < 0.05 for paternal safe haven support) and better overall health and well-being children tend to experience (the correlation coefficient varies between r = 0.42, *p* < 0.05 for maternal secure base support and r = 4.72, *p* < 0.05 for paternal secure base support). In turn, children who experience lower sleep quality tend to experience worst general health and well-being (r = −0.45, *p* < 0.05). 

### 3.3. Mediation Models

We tested four mediation models, as well as direct effects, using PROCESS macro, for the role of sleep quality as a mediator of the effects in the relation between attachment (each one of the four models included one of the attachment dimensions, as in Figure 1) and child well-being. In order to test the significance of the indirect effects, bootstrapping was used; around 2000 bootstrapped samples were used to increase statistical power, with 95% bias-corrected confidence intervals using the maximum likelihood function.

#### 3.3.1. Secure Base Support (Mother), Sleep, and Well-Being

The first model (Table 3), testing for sleep quality as a mediator of the links between maternal secure base support and child-perceived well-being, was significant [F (2, 255) = 57.11, *p* < 0.001, R^2^ = 0.31]. Findings indicated that secure base support of the attachment relationship with the mother had a direct positive association with a sense of well-being. As expected, there was also a significant indirect association between secure attachment and well-being. Direct effects show support for a significant pathway from secure base support from the mother to sleep quality (β = −0.10, *p* < 0.001), and sleep quality to overall well-being (β = −0.67, *p* < 0.001). Attachment security accounted for 5% of the variance in sleep quality, and attachment security and sleep quality explained 31% of the variance in overall well-being. The indirect effect remained significant after the inclusion of the mediator in the model, indicating partial mediation (Table 4).

#### 3.3.2. Safe Haven Support (Mother), Sleep, and Well-Being

The second model suggested that sleep quality significantly mediates the links between the safe haven support function of the attachment relationship with the mother and overall well-being [F (2, 255) = 58.64, *p* < 0.001, R^2^ = 0.32] (Table 5). A direct effect of safe-haven support from the mother on well-being (β = 0.32, *p* < 0.001), as well as an indirect effect via sleep quality, were observed (Table 6), suggesting partial mediation. We also found significant direct associations between maternal safe haven support and sleep (β = −0.16, *p* < 0.001) and between sleep and well-being (β = −0.60, *p* < 0.001). Attachment safe-haven support from the mother accounted for 10% of the variance in sleep quality, and attachment and sleep together explained 32% of the variance in well-being.

#### 3.3.3. Secure Base Support (Father), Sleep, and Well-Being

Regarding attachment to the father, the model testing for the mediation role of sleep quality in the relation between paternal secure base support and well-being was significant [F (2, 252) = 60.29, *p* < 0.001, R^2^ = 0.32] (Table 7) and the indirect effect was indicative of partial mediation (Table 8). Direct effects suggest a significant path from paternal secure base support to sleep quality (β = −0.12, *p* < 0.001) and from sleep quality to well-being (β = −0.58, *p* < 0.001). Secure base support from the father explained 8% of the variance in sleep quality, and together with sleep quality, accounted for 32.4% of the variance in overall well-being.

#### 3.3.4. Safe Haven Support (Father), Sleep, and Well-Being

Finally, similar to findings for a secure base, the last model (Table 9) that examined sleep quality as a mediator of the link between paternal safe haven support and child-perceived well-being was significant [F (2, 252) = 51.92, *p* < 0.001, R^2^ = 0.29]. The safe haven function of the attachment relationship with the father also had a direct effect on well-being, such that greater safe-haven support was associated with a greater sense of well-being. Safe haven support from the father explained 12% of the variance in sleep quality, and together with sleep quality, accounted for 29% of the variance in overall well-being. Both the direct effects of paternal safe haven support in sleep and of sleep in well-being were significant (β = −0.14, *p* < 0.001, and β = −0.56, *p* < 0.001, respectively), as well as the indirect effect, pointing to partial mediation (Table 10).

## 4. Discussion

The present study aimed to examine a bioregulatory system—sleep—through which children’s perceptions of their attachment to their mothers and fathers predict their overall sense of well-being. We hypothesized that sleep quality would mediate the association between the dimensions of safe haven and secure base support in the attachment relationship with the parents and children’s overall well-being. Our results supported the hypotheses establishing that the relationship between attachment to both parent and child’s subjective perception of well-being is partially mediated by sleep problems. The effects of attachment security on well-being, directly and via its impact on sleep, will be discussed, considering the attachment relationship to mother and father as both a secure base from which the child can confidently explore the world and receive a haven of safety in times of distress.

### 4.1. Attachment Functions and Figures

Results show that children who perceive one of the parents as a source of secure base support, encouraging and supporting exploration and novelty tend to perceive the same parent as a safe haven in times of need, being available and responsive to the child. The relatedness of these two components is in line with attachment theory, which defines attachment as a relationship that both facilitates exploration of the environment and comforts the child when distressed [77]. Thus, the two aspects of attachment representation are expected to be related [75,113,114,115], as recent studies point out [77,112], considering they are “two sides of the same attachment coin” [116] (p. 2). According to the mentioned perspective, children who are better able to use attachment figures as a secure base for exploring the world and as a safe haven of comfort in times of distress are thought to possess secure internal representations of attachment.

Besides, children who report obtaining a higher secure base and safe haven support from the mother tend to see the father as a source of secure base and safe haven support, too, suggesting that both mothers and fathers continue to serve as attachment figures in late middle childhood [75,114] and that the child’s working model of attachment is somewhat consistent across relationships. Previous research had already pointed in this direction, indicating that attachment to mother is not independent of attachment to father [70,117], meaning that, in terms of attachment functions, children tend to perceive mother and father in a similar way, although not necessarily identical.

Although secure base and safe haven support from mother and father were moderately correlated, some significant differences were also found. Mothers obtained higher attachment security scores than the fathers, indicating that security levels in the attachment relationship with the mother tend to be higher than with the father. Previous results are mixed. Some also suggest that attachment to female attachment figures is of higher quality than to male attachment figures [118,119]. While mothers in late middle childhood are still the main attachment figure for girls and boys, we found that boys classify their attachment to the mother with higher security than girls. Even though sex differences are not contemplated by attachment theory [120,121], some studies suggest that the emergence of sex differences in late middle childhood is possibly related to the reorganization of the endocrine mechanisms that impact brain development, triggering sex-specific psychological trajectories [122,123]. Other authors [30,124,125] report a decrease in closeness and an increase in conflict and in emotional distance between parents and children as they approach adolescence. Given that girls approach preadolescence earlier than boys, showing more precocious signs of de-idealization of the figures of primary identification [126], they may, in consequence, see their mothers as more distant emotionally than the boys. Despite the differences between girls and boys regarding attachment to the mother, they seem to see the father very similarly. The fathers have been described more as a source of a secure base than of safe-haven support. This is in line with previous findings, indicating that the child’s explorative behavior may be particularly encouraged within the relationship with the father [75,77], in which children are exposed to more challenging, risk-taking games and activities [127,128]. Given that exploration and expansion of the social world are the main tasks of late middle childhood and preadolescence, father attachment may become particularly salient during this phase.

### 4.2. Attachment and Well-Being

Our results show that both the dimensions of attachment relationships to mother and father correlate significantly with overall well-being. Attachment theory has been shown to be a powerful framework in the prediction of child well-being, with previous studies reporting consistent results with socioemotional adaptation [16], self-regulation [58], psychological well-being [129], and life satisfaction [130]. One of the mechanisms involved in this association concerns the links between attachment and self-confidence, predicting that securely attached children who have strong, secure base support tend to be more participative and active in facing middle childhood challenges and activities, which makes them feel happy and satisfied. Moreover, securely attached children, who experience strong, safe haven support, may be more effective in activating the stress response, seeking help and feeling comforted in stressful situations, resulting in an increase in overall well-being.

### 4.3. The Role of Sleep

Our study also added understanding about how the secure base and safe haven functions of the attachment relationships with mother and father impact well-being through their effects on sleep quality. We found evidence for a partial mediation of sleep quality, suggesting that attachment relationships with caregivers are connected to sleep quality, which can ultimately explain the link with well-being. The present study introduces an exploration of preadolescents’ sleep, with sleep quality perceptions assessed through a self-report, as a mechanism through which attachment to mother and father (secure base and safe haven support) impacts the child’s overall well-being. Specifically, the more the child perceives her relationship with their parents as a source of secure base and safe haven support, the fewer sleep problems she is likely to experience, which, in turn, translates into higher self-reported subjective well-being.

Research has previously suggested that attachment security relates to maternal reports of longer nocturnal wakings [109], actigraphy sleep durations and efficiency in toddlers [131], and actigraphy indices of sleep quality in preschoolers [132]. In turn, children with insecure attachment relationships tended to sleep more poorly [133], waking up more often during sleep [134] in infancy. Particularly, children with disorganized attachments show shorter sleep durations, later bedtimes, and longer night-wakings, as reported by the mothers [135]. Although most studies used samples of younger children (between 6 months and 5 years), some research findings have been reported for middle childhood, suggesting negative associations between child attachment to mother and self-reported sleep problems in children aged between 8 and 9 years old [110]. The links between sleep quality and child well-being have also found empirical support in previous studies, relating shorter sleep durations to lower health-related quality of life [101,136], higher odds of externalizing behavior [85], depression [137], and mental health problems [138,139]. In turn, appropriate sleep durations associated with delayed school start times predicted fewer depressive symptoms, lower levels of daytime sleepiness, and less negative mood [140].

### 4.4. Limitations and Future Directions

Our study has some limitations that call for careful interpretations of the results. First, as most of the participants were raised by both parents, generally well-educated and full-time workers, living in the metropolis and its surroundings, we cannot be sure that these results generalize to populations from more diverse backgrounds. Therefore, future studies should include participants with alternative household configurations and different socioeconomic environments. Second, the cross-sectional nature of our findings does not allow us to draw conclusions about the potential underlying causal processes of the associations. Although we have a strong theoretical framework suggesting the plausibility of the discussed direction of the results, future studies should consider longitudinal designs to explore the direction of the associations. Third, the exclusive use of self-report measures to assess attachment security, sleep quality, and well-being may boost positive results due to shared method variance. Despite being widely accepted and reliable compared to others’ reports, middle school-aged children’s self-reports should be used in addition to experimentally-based, observational and interview measures. Last, there are other factors that could also mediate the relationship between attachment security and well-being, such as family sleep routines and marital conflict, should be investigated.

The limitations should be considered in the context of the strengths and implications of our work. We focused on a global construct of well-being that is believed to be adequate in late middle childhood. However, it would be interesting to investigate to what extent the attachment relationships with mother and father play different roles in the development of distinct aspects of well-being. Our study considers the mediating role of sleep quality in the association between attachment and well-being. Taking both dimensions of attachment to mother and father into consideration emphasizes not only the importance of including both figures in research designs but also in therapeutic interventions for children with problems associated with sleep and well-being. Finally, understanding how sleep is related to well-being and attachment relationships in late childhood is of clinical relevance since difficulties associated with sleep can be a precocious and sensitive marker to help parents and health professionals to identify emerging physical and mental health problems.

## 5. Conclusions

The present work focuses on the relationship between attachment, sleep, and well-being in an understudied period of developmental change: late middle childhood. This period marks the transition to adolescence and translates into important challenges to attachment relationships, sleep regulation and well-being. As the child’s social world expands and the wishes for autonomy arise, child–parent dyads may have to renegotiate the mutually acceptable levels of proximity which may impact the attachment system [77]. Regarding sleep regulation, delayed bedtimes [141], along with less parental control of bedtime routines, can result in shorter sleep durations [142]. Besides, children face important vulnerabilities [22] that may reflect reduced subjective well-being [23,24,25]. As such, our study analyzed the relationship between attachment to both parental figures, perceptions of sleep quality, and well-being. Besides the positive relations between attachment security, sleep quality and subjective well-being, we found evidence for an indirect effect of attachment on well-being via sleep quality. Thus, our research adds to knowledge about how relational self-regulatory processes (attachment to both parents) impact physiology-related processes, such as sleep quality, in the prediction of well-being.

## Figures and Tables

**Figure 1 ijerph-20-03399-f001:**
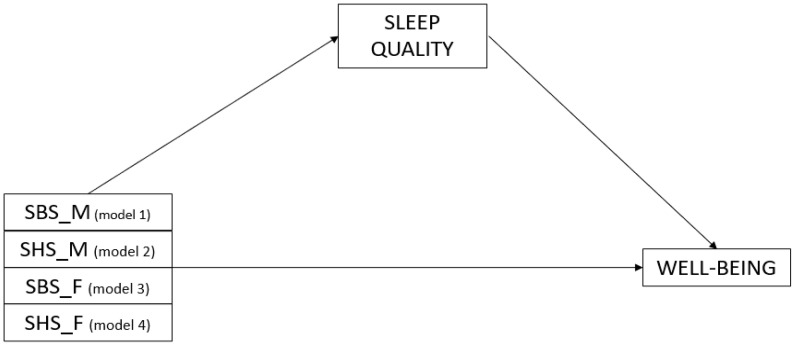
Representation of the four mediation models with the designations of the variables; SBS_M: secure base support (mother); SHS_M safe haven support (mother); SBS_F: secure base support (father); SHS_F safe haven support (mother).

**Table 1 ijerph-20-03399-t001:** Mean scores and standard deviations (in brackets) obtained in the total sample for girls, boys, and for attachment dimensions, sleep quality, and overall well-being.

Variable	Total	Girls	Boys	*p* for the Differences between Boys and Girls (*t*-Test)
SBS_M	3.47 (0.47)	3.35 (0.04)	3.58 (0.04)	>0.01
SHS_M	3.44 (0.45)	3.35 (0.04)	3.51 (0.04)	>0.01
SBS_F	3.38 (0.54)	3.32 (0.05)	3.43 (0.05)	0.10
SHS_F	3.22 (0.58)	3.22 (0.05)	3.21 (0.05)	0.91
Sleep quality	1.55 (0.22)	1.56 (0.23)	1.53 (0.22)	0.30
Well-being	4.07 (0.40)	4.03 (0.04)	4.11 (0.04)	0.07

SBS_M: Maternal secure base support; SHS_M: Maternal safe haven support; SBS_F: Paternal secure base support; SHS_F: Paternal safe-haven support.

**Table 2 ijerph-20-03399-t002:** Partial correlations between attachment dimensions, sleep, and well-being.

	1.	2.	3.	4.	5.	6.
1. SBS_M	1					
2. SHS_M	0.64 **	1				
3. SBS_F	0.58 **	0.55 **	1			
4. SHS_F	0.40 **	0.61 **	0.69 **	1		
5. Sleep	−0.21 **	−0.31 **	−0.28 **	−0.35 **	1	
6. Well-being	0.42 **	0.46 **	0.47 **	0.45 **	−0.45 **	1

** *p* < 0.05. SBS_M: maternal secure base support; SHS_M: maternal safe haven support; SBS_F: paternal secure base support; SHS_F: paternal safe-haven support.

**Table 3 ijerph-20-03399-t003:** Mediation effect of the sleep quality in the link between secure base support from the mother (SBS_M) and overall well-being.

		Consequent						
		M (Sleep)				γ (CHIP-CE)		
Antecedent		Coeff.	SE	*p*		Coeff.	SE	*p*
X (SBS_M)	a	−0.10	0.03	<0.001	c	0.29	0.05	<0.001
M (Sleep)		-	-	-	b	−0.67	0.09	<0.001
constant	i_M_	1.90	0.10	<0.001	i_γ_	4.10	0.24	<0.001
		R^2^ *=* 0.05				R^2^ = 0.31		
		F (1, 256) = 12.17, *p* < 0.001				F (2, 255) = 57.11, *p* < 0.001		

Indirect effect = ab = −0.10 × −0.67 = 0.07.

**Table 4 ijerph-20-03399-t004:** Indirect effect of the secure base support from the mother (SBS_M) on sleep quality.

	Effect	BootSE	BootLLCI	BootULCI
Sleep	0.07	0.02	0.03	0.12

**Table 5 ijerph-20-03399-t005:** Mediation effect of the sleep quality in the link between safe-haven support from the mother (SHS_M) and overall well-being.

		Consequent						
		M (Sleep)				γ (CHIP-CE)		
Antecedent		Coeff.	SE	*p*		Coeff.	SE	*p*
X (SHS_M)	a	−0.16	0.03	<0.001	c	0.32	0.05	<0.001
M (Sleep)		-	-	-	b	−0.60	0.10	<0.001
constant	i_M_	2.09	0.11	<0.001	i_γ_	3.90	0.26	<0.001
		R^2^ = 0.10				R^2^ = 0.32		
		F (1, 256) = 26.88, *p* < 0.001				F (2, 255) = 58.64, *p* < 0.001		

Indirect effect = −0.16 × −0.60 = 0.10.

**Table 6 ijerph-20-03399-t006:** Indirect effect of the safe haven support from the mother (SHS_M) on sleep quality.

	Effect	BootSE	BootLLCI	BootULCI
Sleep	0.10	0.03	0.05	0.15

**Table 7 ijerph-20-03399-t007:** Mediation effect of the sleep quality in the link between secure base support from the father (SBS_F) and overall well-being.

		Consequent						
		M (Sleep)				γ (CHIP-CE)		
Antecedent		Coeff.	SE	*p*		Coeff.	SE	*p*
X (SBS_F)	a	−0.12	0.025	<0.001	c	0.28	0.04	<0.001
M (Sleep)		-	-	-	b	−0.58	0.10	<0.001
constant	i_M_	1.94	0.085	<0.001	i_γ_	4.04	0.22	<0.001
		R^2^ = 0.08				R^2^ = 0.32		
		F (1, 253) = 21.42, *p* < 0.001				F (2, 252) = 60.29, *p* < 0.001		

Indirect effect: −0.12 × −0.58 = 0.07.

**Table 8 ijerph-20-03399-t008:** Indirect effect of the secure base support from the father (SBS_F) on sleep quality.

	Effect	BootSE	BootLLCI	BootULCI
Sleep	0.07	0.02	0.03	0.11

**Table 9 ijerph-20-03399-t009:** Mediation effect of the sleep quality in the link between safe-haven support from the father (SHS_F) and overall well-being.

		Consequent						
		M (Sleep)				γ (CHIP-CE)		
Antecedent		Coeff.	SE	*p*		Coeff.	SE	*p*
X (SHS_F)	a	−0.13	0.02	<0.001	c	0.23	0.04	<0.001
M (Sleep)		-	-	-	b	−0.56	0.10	<0.001
constant	i_M_	1.99	0.08	<0.001	i_γ_	4.18	0.23	<0.001
		R^2^ = 0.12				R^2^ = 0.29		
		F (1, 253) = 35.51, *p* < 0.001				F (2, 252) = 51.92, *p* < 0.001		

Indirect effect = 0.08.

**Table 10 ijerph-20-03399-t010:** Indirect effect of the safe haven support from the father (SHS_F) on sleep quality.

	Effect	BootSE	BootLLCI	BootULCI
Sleep	0.077	0.019	0.043	0.118

## Data Availability

The raw data supporting the conclusions of this article will be made available by the authors without undue reservation.

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
