# Peer review of "Attachment to Mother and Father, Sleep, and Well-Being in Late Middle Childhood"

_ijerph, 2023, doi:10.3390/ijerph20043399_

Round 1

Reviewer 1 Report

Attachment to mother and father, sleep, and well-being in late middle childhood

The article explores important yet understudied topic in the adolescents literature. I enjoyed reading this carefully constructed paper. My comments are below.

Abstract

Abstract is well written and contains all essential info. Perhaps re-write this sentence: “Attachment relationship has been related to several developmental outcomes, such as the absence of sleep problems and diverse facets of well-being.” to add some clarity – do you mean secure attachment has been found related to advantageous development outcomes such as good sleep and higher wellbeing measures?

Introduction

Can you shorten this sentence to add some clarity: “However, across the variety of suggested child well-[36] being definitions, White [11] noted some consistency regarding a positive approach (e.g., 37 capacity for experiencing positive emotions, positive interactions with others, social, and 38

academic adaptation; [12, 13]; and absence of negative affect or symptoms of mental dis- 39

tress; [14, 15]), a holistic emphasis considering the interaction of the different elements 40

and the value attributed to the subjective perceptions.”

In general, I am not sure what is the focus of the first two paragraphs of the introduction. Perhaps it could be shorted so the introduction is more focused on the actual literature directly relevant to this study.

Page 3, lines 138-140: Specifically, longer sleep duration and better sleep quality are 138

associated with higher levels of well-being, measured as psychological attitudes towards 139

life, future, recent events, family life and friendships, both cross-sectionally and 6 months 140

later [92, 93]. Please change long sleep for appropriate for a given age sleep during; long sleep in the literature is primarily used to indicate too long/unhealthy durations. Please do so in the entire paper (e.g. Discussion, p.10, l. 435).

Method

Page 4, line 183 – is this a commonly used term “intact families”?

Table 1 -explain that vales are means and SD. Also, can you add p-vales to this table?

Table 2 – explain again in the table legend what the abbreviations used mean (like you did for Table 1).

Results

Mediation models – please justify why bootstrapping was at 2000. This is not the default number.

I find it odd that results for mediation analyses as shown only as supplementary tables (and this is not mentioned in the Results when the reader is directed to a particular table, but only at the end of the Discussion). I appreciate that each journal has a limit of the number of tables that are shown in the main article, but perhaps you can combine your tables in one or two? It is much easier for the readers to look at tables than to read the text to see what the results are.

Discussion

Page 8 and 9 lines 370-374, you write that “attachment to mother is not independent from attachment to father (…) children tend to perceive mother and father in a similar way, although not necessarily identical.” -does this apply to “traditional” families when both parents are present? How about families where parents are divorced, etc?

Some grammatical errors, e.g. p9, l.413: @We found evidence for a partial mediation of sleep quality, suggesting that attachment relationships with the caregivers is connected TO sleep quality

p.11, l.473-4 “Regarding sleep regulation, the delayed bedtimes along with a less parental control of bedtime routines, CAN RESULT reflect in shorter sleep durations” .

Author Response

We sincerely thank the Reviewers for their insightful and constructive feedback on our manuscript. Replies to all comments and suggestions are given in the following pages, and the corresponding changes in the manuscript are marked by the “Track Changes” tool. We confide that the comments and the ensuing revisions made the manuscript a stronger contribution to the field.

Reviewer: 1

  1. Abstract is well written and contains all essential info. Perhaps re-write this sentence: “Attachment relationship has been related to several developmental outcomes, such as the absence of sleep problems and diverse facets of well-being.”

To add some clarity – do you mean secure attachment has been found related to advantageous development outcomes such as good sleep and higher wellbeing measures?

We appreciate the Reviewer’s suggestion to add some clarity and we rewrote the sentence in order to focus on the relations between attachment security and better developmental outcomes.

  1. Can you shorten this sentence to add some clarity: “However, across the variety of suggested child well-[36] being definitions, White [11] noted some consistency regarding a positive approach (e.g., 37 capacity for experiencing positive emotions, positive interactions with others, social, and 38

We thank R#1 for this careful revision. We reviewed the sentence, uncovering the conceptualization of well-being in terms of a holistic construct.

  1. In general, I am not sure what is the focus of the first two paragraphs of the introduction. Perhaps it could be shorted so the introduction is more focused on the actual literature directly relevant to this study.

We found it important to go lightly through the history of child well-being conceptualizations since it can sometimes be difficult to understand what authors refer to when they include child well-being in their studies. We found that the presentation of a theoretical background supporting the conceptualization of well-being could be important from the beginning fulfilling purposes of theoretical integration since in other studies we would only find clarification in the instruments section. That is the reason why the first paragraph is dedicated to the definition of well-being in the presented study. The second one intends to justify why the study focused on preadolescent years, an age range marked by several changes that impact subjective perceptions of well-being.

  1. Page 3, lines 138-140: Specifically, longer sleep duration and better sleep quality are 138

associated with higher levels of well-being, measured as psychological attitudes towards 139

life, future, recent events, family life and friendships, both cross-sectionally and 6 months 140

later [92, 93]. Please change long sleep for appropriate for a given age sleep during; long sleep in the literature is primarily used to indicate too long/unhealthy durations. Please do so in the entire paper (e.g. Discussion, p.10, l. 435).

We totally agree with this suggestion, and we changed long sleep for age-appropriate or appropriate sleep in the entire paper.

5 Page 4, line 183 – is this a commonly used term “intact families”?

            “Intact family” is used to designate a married or common-law couple where all the children are offspring of both members of the couple. However, we replaced this term with its definition to add clarity.

  1. Table 1 -explain that vales are means and SD. Also, can you add p-vales to this table?

            We added some information to the table, regarding what we are presenting (Means and SD’s). We also added p-values to this table.

  1. Table 2 – explain again in the table legend what the abbreviations used mean (like you did for Table 1).

            We explained the meaning of the abbreviations, like we did for Table 1.

  1. Mediation models – please justify why bootstrapping was at 2000. This is not the default number.

            However 2000 is not the default number, we considered it to be a safer value to increase statistical power, given the limited size of our sample.

  1. I find it odd that results for mediation analyses as shown only as supplementary tables (and this is not mentioned in the Results when the reader is directed to a particular table, but only at the end of the Discussion). I appreciate that each journal has a limit of the number of tables that are shown in the main article, but perhaps you can combine your tables in one or two? It is much easier for the readers to look at tables than to read the text to see what the results are.

            We added the tables that previously integrated the Supplementary Materials to the main text.

  1. Page 8 and 9 lines 370-374, you write that “attachment to mother is not independent from attachment to father (…) children tend to perceive mother and father in a similar way, although not necessarily identical.” -does this apply to “traditional” families when both parents are present? How about families where parents are divorced, etc?

            This is an excellent question, however, in this study, we focused on “traditional” when both parents are present, unfortunately we do not have data on the reality of other familial configurations. This should be taken in consideration in future studies.

  1. Some grammatical errors, e.g. p9, l.413: @We found evidence for a partial mediation of sleep quality, suggesting that attachment relationships with the caregivers is connected TO sleep quality, p.11, l.473-4 “Regarding sleep regulation, the delayed bedtimes along with a less parental control of bedtime routines, CAN RESULT reflect in shorter sleep durations”.

            We thank the reviewer for noticing some grammatical mistakes and we corrected them.

Reviewer 2 Report

The Authors present a mediation study between the variables of attachment to mother and father, sleep and well-being during late middle chidlhood.
Overall, the study is well written and documented.

INTRODUCTION
The introduction is accompanied by appropriate and up-to-date citations, and the reading is fluent. One small note I would like to make (line 62 and following) is the possibility of interpreting attachment not only as trait-like characteristics in an individual's personality but also as the possibility of multiple attachments and related multiple mental patterns, thus as context-specific attachment patterns.
A valuable study is the following, focusing on young adults:
Caron et al. (2012). Comparisons of close relationships: An evaluation of relationship quality and patterns of attachment to parents, friends, and romantic partners in young adults.
Another aspect that the Authors might consider are the dimensions of attachment, specifically anxiety and avoidance. Thus, not only attachment styles and attachment security or insecurity, but also in the two orthogonal dimensions proposed by Bartholomew and Horowitz (1991) (*Attachment styles among young adults: A test of a four-category model).

Line 163: typo ("stud")

MATERIALS AND METHODS
2.1 Participants: it may be useful to have performed a power analysis to estimate sample size or, alternatively, a mention to sample size that is sufficient for analyses. In addition, it might be useful to expand the section by describing in more detail the reasons for exclusion (i.e., going from 477 to 258 participants) (lines 180-182).

2.3 Instruments and 3. Results
Lines 255-256: The distinction between secure attachment and the two dimensions measured by Kerns Security Scale is not clear. Is it assumed that participants' attachment is always of the "secure" type when they score above a threshold on the two subscales of the Kerns Security Scale? This point should be expressed in more detail in the 2. Materials and Methods section.

3.2 Partial correlations (lines 275-289): it is plausible that the two dimensions measured by the Kerns Security Scale are correlated with each other. Has the internal consistency of the instrument been measured?

Overall, the Results sections could be improved, with clear tables and a more straightforward presentation.

4. Discussion (line 346 and following).
The authors in both their presentation of results and discussion consider the two dimensions assessed by the Kerns Security Scale to be related to secure attachment. This connection is not clear. It would have been appropriate to assess the type of attachment with a more specific instrument to go along with the Kerns Security Scale. This aspect could be added in the Limits section.

Author Response

  1. The introduction is accompanied by appropriate and up-to-date citations, and the reading is fluent. One small note I would like to make (line 62 and following) is the possibility of interpreting attachment not only as trait-like characteristics in an individual's personality but also as the possibility of multiple attachments and related multiple mental patterns, thus as context-specific attachment patterns. A valuable study is the following, focusing on young adults: Caron et al. (2012). Comparisons of close relationships: An evaluation of relationship quality and patterns of attachment to parents, friends, and romantic partners in young adults.

            We thank the reviewer for bringing the theoretical background of attachment theory into the discussion. As our study focuses on the representations of two distinct attachment relationships (i.e., mother and father), we strongly agree that attachment security is a relationship-specific characteristic – particularly at such a young age – and less a trait-like characteristic. Our results somehow support this claim, since the correlations between attachment to both parents are only moderate. We reviewed some literature regarding the differences between attachment to mother and father across the development, but we did not deepen the theoretical debate considering the monotropism versus hierarchy of attachment representations.

Another aspect that the Authors might consider are the dimensions of attachment, specifically anxiety and avoidance. Thus, not only attachment styles and attachment security or insecurity, but also in the two orthogonal dimensions proposed by Bartholomew and Horowitz (1991) (*Attachment styles among young adults: A test of a four-category model).

We do appreciate your comment, and we consider  typologica approach an important and valued one. However, our approach in the present article is more continuous and focus on  how the child uses the attachment figure as a secure base (secure base support function) from which to explore the world and as a haven of safety to regain emotional reassurance, should some threat arise (safe haven support function). We did not assess attachment styles and attachment security or insecurity in a typological fashion, but rather in assessing through a continuous variable the extent to which a child can rely on her attachment figures with their primary functions of encouraging autonomy (secure base) and comforting emotionally (safe haven). This choice articulates with the nature of the developmental phenomena at play during late middle childhood, a time when the child boosts even more the exploration of the world, while still being dependent on the caregivers. 

  1. Participants: it may be useful to have performed a power analysis to estimate sample size or, alternatively, a mention to sample size that is sufficient for analyses. In addition, it might be useful to expand the section by describing in more detail the reasons for exclusion (i.e., going from 477 to 258 participants) (lines 180-182).

            We added to the reasons for exclusion.

  1. Lines 255-256: The distinction between secure attachment and the two dimensions measured by Kerns Security Scale is not clear. Is it assumed that participants' attachment is always of the "secure" type when they score above a threshold on the two subscales of the Kerns Security Scale? This point should be expressed in more detail in the 2. Materials and Methods section.

            We reformulated this section to clarify how the Kerns Security Scale assesses attachment security in two different dimensions.

  1. Partial correlations (lines 275-289): it is plausible that the two dimensions measured by the Kerns Security Scale are correlated with each other. Has the internal consistency of the instrument been measured?

We had mentioned the interval where the α’s were included, but now we specified the Cronbach- α’s as measures of internal consistency for each subscale of the instrument.

  1. Overall, the Results sections could be improved, with clear tables and a more straightforward presentation.

            We improved the Results sections by inserting the Tables along with the text.

  1. The authors in both their presentation of results and discussion consider the two dimensions assessed by the Kerns Security Scale to be related to secure attachment. This connection is not clear. It would have been appropriate to assess the type of attachment with a more specific instrument to go along with the Kerns Security Scale. This aspect could be added in the Limits section.

We clarified the association between attachment security and the dimensions of Secure Base and Safe Haven support.